# Clinical Safety Assessment of Autologous Freeze-Drying Platelet-Rich Plasma for Bone Regeneration in Maxillary Sinus Floor Augmentation: A Pilot Study

**DOI:** 10.3390/jcm10081678

**Published:** 2021-04-14

**Authors:** Takamitsu Koga, Yuya Nakatani, Seigo Ohba, Masahito Hara, Yoshinori Sumita, Kazuhiro Nagai, Izumi Asahina

**Affiliations:** 1Department of Regenerative Oral Surgery, Institute of Biomedical Sciences, Nagasaki University Graduate School of Biomedical Sciences, 1-7-1 Sakamoto, Nagasaki 852-8588, Japan; kogataka215@gmail.com (T.K.); y.naka.dds.2011@gmail.com (Y.N.); seigoohba@gmail.com (S.O.); hara.m@nagasaki-u.ac.jp (M.H.); 2Department of Dentistry and Oral Surgery, Imaki-ire General Hospital, Kagoshima 892-8502, Japan; 3Basic and Translational Research Center for Hard Tissue Disease, Nagasaki University Graduate School of Biomedical Sciences, 1-7-1 Sakamoto, Nagasaki 852-8588, Japan; y-sumita@nagasaki-u.ac.jp; 4Transfusion and Cell Therapy Unit, Nagasaki University Hospital, 1-7-1 Sakamoto, Nagasaki 852-8501, Japan; agwkn@nagasaki-u.ac.jp

**Keywords:** freeze-drying platelet-rich plasma, platelet-rich plasma, sinus floor augmentation, bone regeneration

## Abstract

The purpose of this clinical study is to evaluate the safety and preliminary efficacy of autologous freeze-drying platelet-rich plasma (FD-PRP) on bone regeneration in maxillary sinus floor augmentation as a preliminary pilot study. Five patients that required sinus floor augmentation to facilitate the placement of dental implants participated in this clinical study. The PRP was prepared from the autologous peripheral blood and was lyophilized and stored at −20 °C for 4 weeks before surgery. At surgery, triple-concentrated FD-PRP (x3FD-PRP) mixed with synthetic bone grafting materials was rehydrated following the transplantation into the sinus floor. The primary outcome was a safety verification of x3FD-PRP, evaluated in terms of the clinical course and consecutive blood tests. The secondary outcome was clinical efficacy focused on bone regeneration in sinus floor augmentation evaluated by radiographic examination and implant stability. There were no adverse events, such as systemic complications, excessive inflammatory reactions, severe infection, or local site healing complications, besides those on the usual course associated with surgery. Vertical augmented height was maintained, and the initial stability of implants was achieved post-operatively in 6 months. The results obtained in this study suggest that x3FD-PRP can be used safely for bone engineering in clinical practice. Further studies are required to draw a conclusion concerning the efficacy of x3FD-PRP since this was a pilot study with a single arm and a small sample size.

## 1. Introduction

Platelet-rich plasma (PRP) is defined as platelets concentrated over the basal number (4- to 9-fold) in a small plasma volume [1] obtained from autologous peripheral blood (PB). PRP includes various growth factors, such as platelet-derived growth factor (PDGF), transforming growth factor (TGF-β), vascular endothelial growth factor (VEGF), fibroblast growth factor (FGF), epithelial growth factor (EGF), and adhesion molecules (i.e., fibrin, fibronectin, and vitronectin). Such growth factors and molecules have been shown to promote cell recruitment, proliferation, and angiogenesis which may be implicated in tissue regeneration and healing [1,2]. Therefore, it has been used for anagenesis treatment in the multi-field.

The recovery of bone deficiency caused by trauma, tumor resection, and aging has been a challenge in the field of oral and maxillofacial surgery. Autogenous bone grafting is still the gold standard for bone augmentation because of its excellent osteoinductivity and osteoconductivity, but it has some impediments such as limited availability and donor site morbidity. Alternative bone grafting materials, including allografts, xenografts, and alloplastic bone grafts, are used clinically, but they do not have osteoinductivity. It is expected that the use of PRP will accelerate osteoinducibility in combination with these bone grafting materials and improve the osteogenic effect. Many clinical studies have shown that PRP is effective in alveolar bone regeneration [3,4]. 

However, there are several issues that still need to be addressed to optimize the clinical use of PRP. PRP is generally isolated and applied on site during surgery to maintain its activity, but it is time- and labor-intensive and it can be difficult to prepare adequate amounts. Thus, it would be useful if PRP could be isolated beforehand and stored until use. We focused on the technique of freeze-drying PRP (FD-PRP), which was first reported by Pietramaggiori et al. [5] to overcome this problem. Some studies have indicated the possibility of FD-PRP to maintain useful growth factor levels in the supernatant [6,7,8]. Our pre-clinical study demonstrated that PRP can be stored without functional loss by freeze-drying and the concentration of FD-PRP may improve its efficacy in bone engineering [9]. On applying FD-PRP in a clinical setting, the risk of complications, such as systemic complications, infection, and excessive inflammatory reactions (e.g., graft-versus-host disease), should be carefully assessed. FD-PRP is expected to be of great value in clinical practice; however, there is little clinical research regarding this area. The purpose of this clinical study is to evaluate the safety and efficacy of triple-concentrated FD-PRP (x3FD-PRP) on bone regeneration in maxillary sinus floor augmentation in a pilot study. 

## 2. Materials and Methods

### 2.1. Design of the Clinical Trial and Patient Selection 

The protocol of the present clinical trial was submitted and approved by the clinical research review board in Nagasaki University (approval no. CRB7180001) and has been registered with the Japan Registry of Clinical Trials (trial no. jRCTs072180076) and the University Hospital Medical Information Network in Japan (register no. JPRN-UMIN000027144) as “Efficacy of freeze-dried platelet-rich plasma (FD-PRP) in bone engineering: Pilot study in bone engineering”. The study was performed according to the Declaration of Helsinki. All patients were fully informed about the procedures and provided written informed consent prior to enrolment. The study type is interventional, single-arm, open, uncontrolled, and single assignment. This study was conducted between May 2017 and September 2020. 

The candidates were enrolled according to the following inclusion criteria and underwent an interview about their underlying health conditions and blood tests before the operation. Five patients who signed the formal consent form were included in this study. These 5 patients included 4 females and 1 male, and the ages of the patients ranged from 39 to 64, with an average age of 52.0 years (Table 1). 

#### 2.1.1. Inclusion Criteria 


Patients who are partially or fully edentulous and are required to be treated with dental implants for prosthetic rehabilitation.Patients who have insufficient bone height and/or width to place the dental implant.Patients who have received oral health care and maintain a good condition of plaque control.The age of the patients: 20 to 75 years old.


#### 2.1.2. Exclusion Criteria 


All patients are required to be non-smokers.Patients who have systemic diseases, malignancies, chronic infections, immune system abnormalities, septicemia, syphilis, HBV, HCV, HTLV-1, HIV, who are pregnant, or have dementia.Patients who have a blood coagulation disorder.Patients who have hepatitis function disorders, metabolic bone disease, or skeletal dysplasia.Patients who need a legal representative.


Figure 1 shows the protocol of this study. Briefly, the autologous PB was collected within 1 month before surgery. The PRP was prepared, freeze-dried, and stored at −20 °C. This process was performed aseptically at a laminar flow cabinet in a cell processing center. At surgery, the x3FD-PRP mixed with the bone grafting materials was rehydrated and was transplanted. A sterility test was performed in the part of the x3FD-PRP that was used for transplant. 

The primary outcome of this clinical pilot study was a safety verification of x3FD-PRP for alveolar bone regeneration. The safety verification was assessed using the clinical course and in consecutive laboratory examinations. The secondary outcome was a clinical evaluation focused on bone regeneration in sinus floor augmentation analyzed by a radiographic examination and an implant stability quotient. 

### 2.2. Preparation of FD-PRP 

The FD-PRP was prepared according to the standard method described by Nakatani et al. [9]. A maximum of 90 mL of autologous PB was collected for preparing the FD-PRP and autologous serum. PRP was prepared from autologous PB using a blood phase separator (Medifuge MF 200; Silfradent, Santa Sofia, Italy). PB was mixed with one tenth the amount of sodium citrate (Citramin; Fuso, Osaka, Japan) and divided into 7 mL plastic tubes (Venoject1 II VP-P070K30, Terumo, Tokyo, Japan). It was then centrifuged for 2 min at 2700 rpm, 4 min at 2400 rpm, 4 min at 2700 rpm, and 3 min at 3000 rpm, continuously. After centrifugation, 1/10 the volume of whole blood from the border of the plasma and red blood cell (RBC) layers were marked with a marker and the upper layer of plasma was discarded, while the lower layer, including platelets, buffy coat, and small amounts of RBC, was collected with a pipette as PRP. The number of white blood cells (WBC), RBC, and platelets in PB and PRP were measured using an automatic hematology analyzer (MEK-6510 Celltac A; Nihon Kohden, Tokyo, Japan).

Ten milliliters of PB, obtained from the same donors, was then divided into glass tubes without sodium citrate. The tubes were warmed at 37 °C until clot retraction for about 40 min. Samples were then centrifuged at 800× *g* for 10 min (LC-122; Tomy, Tokyo, Japan). Subsequently, the supernatants were collected as autologous serum and were lyophilized and this was used as the thrombin recombinant substitute for PRP activation. 

The PRP or serum was pre-frozen at −80 °C for 12 h and was then lyophilized for 12 h using a lyophilizer (EYELA FD-1000; Tokyo Rikakikai, Tokyo, Japan). After lyophilization, the samples were stored at −20 °C for 1 month in order to prevent contamination and transformation. 

### 2.3. Preparation of the Complex of x3FD-PRP and Bone Graft Materials

The same amount of rehydrated FD-serum and 2% CaCl2 (Otsuka Pharmaceutical Factory, Tokushima, Japan) were mixed as the PRP activator. One unit of x3FD-PRP, which was prepared from 3 mL of PRP, was mixed with 1 g per 1 unit of either β-TCP granules (Cerasorb M; Curasan, NC, USA) or carbonate apatite granules (Cytrans Granules; GC Corporation, Tokyo, Japan) and was rehydrated with 1 ml of sterile purified water per 1 unit on the glass dish. This was then activated with 0.2 mL of PRP activator per 1 unit. After confirming gelatinization, the complex of x3FD-PRP and the β-TCP granules were transplanted immediately. A sterility test was performed in part of the x3FD-PRP which was used for transplantation. Figure 2 shows the preparation process for the complex of x3FD-PRP and bone grafting materials. 

### 2.4. Surgical Procedure 

Maxillary sinus floor augmentation was performed using the lateral window approach as shown in Figure 3. After elevating the mucoperiosteal flap, a bone window was made using a piezoelectric instrument and the maxillary sinus mucous membrane was elevated. After the complex of x3FD-PRP and the bone graft materials were transplanted, the bone window was repositioned and the mucoperiosteal flap was sutured. In case perforation of the maxillary sinus mucosa occurred during surgery, the perforation was closed using absorbent collagen vulnerary covering material when the perforation was less than 5 mm. 

### 2.5. Clinical and Laboratory Assessment 

The safety verification was assessed using the clinical course and consecutive blood tests. The clinical course was evaluated using the following: (1) infection; (2) pain; (3) bleeding; (4) swelling; (5) fever; (6) wound healing; (7) dysesthesia; (8) other adverse events at the four time-points of day. This was checked at 1, 2, and 4 weeks post-operatively. The blood tests were evaluated using the following: a hematological test, biochemical test, and serological test (preoperative assessment) both before and 4 weeks after surgery. 

### 2.6. Radiographic Assessment and Implant Stability

The evaluation focused on bone regeneration in sinus floor augmentation was performed using panoramic radiography and computed tomography (CT). The vertical bone height was measured based on panoramic findings both before and post-operatively on the first day, 1 month, and 6 months. The maximum height of the radiopaque area of the transplanted site was measured as the vertical bone height. The state of maxillary sinusitis or other complications regarding the bone graft site (e.g., bone resorption and leakage of bone grafting materials into the sinus) was confirmed using CT findings at 6 months after surgery. The implant stability was evaluated using the implant survival and implant stability quotient (ISQ) levels of the implant at 6 months post-operatively in the simultaneous approach group using the Osstell ISQ Scale (Osstell, Gothenburg, Sweden). ISQ is known to offer an index of implant stability. The minimum value of ISQ is 1 and the maximum is 100. The implant stability is low when the ISQ value is <60, moderate when between 60 and 70, and high when >70.

### 2.7. Statistical Analysis

Statistical evaluations of blood cell counts and vertical bone heights in each treatment period were performed using a paired *t*-test. These analyses were performed using a software package (JMP® Pro13.0, SAS Institute Inc., Cary, NC, USA). Statistical significance was set at *p* < 0.05. 

## 3. Results

### 3.1. Blood Cell Counts

Table 2 shows the blood cell counts in PB and PRP. The number of WBCs and platelets in PRP were higher than that of the PB by a statistically significant difference. The concentration rates of WBCs and platelets were an average of 3.2-fold and 5.4-fold, respectively, whereas RBCs showed a low level of PRP compared to PB. 

### 3.2. Sterility Test of FD-PRP

Bacteria, mycoplasma, and endotoxin were not detected in the x3FD-PRP used for transplant in all cases.

### 3.3. Clinical and Laboratory Assessment

The following findings were obtained by the clinical evaluation: (1) there were no local-site infections with drainage; (2) there were no severe pains that could not be controlled using an anti-inflammatory analgesic in more than a day; (3) there were no local site bleeds that required hemostasis treatment; (4) there was no severe swelling with dysfunction; (5) there were no fevers over 38 °C after more than a day; (6) there were no wound-healing complications; (7) there were no sensory disturbances; (8) there were no adverse reactions associated with the treatment.

In the blood tests, no serious liver dysfunctions, renal dysfunctions, or electrolyte imbalances were recognized without transient, slight deviations from the normal range at 4 weeks post-operatively.

### 3.4. Radiographic Assessment and Implant Stability

The vertical bone height at the transplanted site was quantitatively measured from panoramic findings (Figure 4). The mean of vertical bone height at the pre-operative stage and post-operative stage at day 1, 1 month, and 6 months was 3.7 ± 1.0, 10.0 ± 3.8, 9.6 ± 3.5, and 9.0 ± 3.6 mm, respectively. A significant difference was observed between the pre-operative stage and each of the post-operative time-points. Evidence of maxillary sinusitis or other complications was not found in all cases from the CT findings of 6 months post-operatively (Figure 5).

In the simultaneous approach group, there were 5 implants in 3 cases. There were no cases in which there was implant loss. The mean ISQ level was 73.6 ± 4.1 with a range of 70–80 at 6 months after surgery. (Table 3)

## 4. Discussion

The first concern of the present study was the safety of x3FD-PRP when used for bone regeneration in maxillary sinus floor augmentation. There were no adverse events such as systemic complications, excessive inflammatory reactions, severe infection, or local site healing complications besides the usual course associated with surgery. Also, vertical augmented height was maintained, and the initial stability of implants was achieved in 6 months post-operatively. Our experimental results may indicate that x3FD-PRP could be applied safely and shows no adverse effects for maxillary sinus floor augmentation.

Treatment using normal fresh PRP for humans has already been provided in clinical practice and has obtained good results with no major complications [1,10]. Because FD-PRP does not use a special reagent for preparation, it is thought that the safety of its use is equivalent to using fresh PRP. However, there are a few reports about clinical studies of FD-PRP that include lyophilized platelet-rich fibrin (Ly-PRF). Morimoto et al. reported that no side effects, such as inflammation or infection, were observed after the application of the FD-PRP during the treatment of a non-healing ulcer on the leg [11]. In addition, Zhang et al. reported that fresh PRF and Ly-PRF mixed with a Bio-Oss bone-graft were applied to guided bone regeneration of the anterior maxillary region and there were no significant differences between flesh PRF and Ly-PRF in factors of histological and clinical evaluations (i.e., color, swelling, bleeding of the mucosa, pain level, and the remodeling of hard tissue) [12]. The present results are consistent with previous studies: no negative effects were identified during the observation period and there were no contaminations detected in the x3FD-PRP. These findings support the safety of freeze-drying and storage processing PRP, as long as the x3FD-PRP is prepared in a clean room using sterile procedures. Since this study is a pilot study with a small number of subjects, it is necessary to conduct a study with a larger number of subjects in order to confirm the safety of x3FD-PRP.

The second concern was to assess the efficacy of applying x3FD-PRP for bone regeneration. The usage of FD-PRP in bone engineering is currently being investigated for a wide range of therapeutic indications across multiple fields in laboratory studies and through animal experimentation. For instance, Shiga et al. reported that FD-PRP maintained baseline levels of growth factor for an entire 8-week duration [6] and that FD-PRP accelerated bone formation with artificial bone used in a rat posterolateral fusion model [13]. Kinoshita et al. reported that PDGF in FD-PRP is pharmacologically active in vitro after 4 weeks of storage [14]. It is also reported that FD-PRP significantly enhanced alkaline phosphatase activity and the mRNA expression level of osteogenic genes in vitro, and also that FD-PRP with scaffolds induced significantly greater bone formation compared to the traditional PRP in a rat calvaria defect [15]. In our previous study, FD-PRP promoted identical levels of bone formation as flesh PRP, and x3FD-PRP induced a more abundant bone formation on a mice calvaria onlay graft model [9]. In the current study, vertical augmented height was maintained for 6 months post-operatively (Figure 4) and initial stability of the implants was achieved (Table 3) without maxillary sinusitis (Figure 5). Specific evaluations of bone regeneration, such as bone biopsy or dense observation with a CT image, were not performed in this study, since the primary outcome of this study was safety verification. Based on previous studies and our results, the freeze-drying and storage process may not exert a negative impact on PRP to promote bone regeneration.

PRP is thought to be effective in tissue regeneration because of its concentrated growth factors: the effectiveness of which can vary depending on the platelet concentration [14]. Marx suggested that a 4- to 9-fold concentration of platelets in the PB, or 100 × 10^4^ platelets/mL, is needed in PRP for tissue healing enhancement [1]. Graziani et al. investigated the effects of different platelet concentrations in PRP using osteoblasts and concluded that a 3- to 5-fold platelet concentration is effective for bone repair [16]. In the current study, an average 5.4-fold of platelet concentration was used for x3FD-PRP preparation (Table 2) and it had a certain therapeutic efficacy. Although the optimal platelet concentration in PRP has not been clarified, the PRP used in this study is thought to be appropriate based on the previous studies.

The benefit of FD-PRP is that it is easy to store as its powder form enables storage in a refrigerator or even at room temperature: this characteristic facilitates the mixture with artificial bone [6]. Furthermore, this characteristic makes it possible to prepare a higher concentration of FD-PRP. Although x3FD-PRP was used in this study in accordance with our previous study, FD-PRP with an increased concentration rate may be effective for bone regeneration. Further studies are necessary to determine the most suitable concentration rate of FD-PRP. There are other advantages such as good operability and an ease of maintenance at the local site because the PRP gelates with the bone grafting materials (Figure 2). It would be useful when perforation occurs in the maxillary sinus mucous membrane.

The present study has several limitations. Firstly, the most critical impediment of this study is the small sample size of five subjects with only one male with a single-arm, even though this was planned to be the first-in-human study to apply x3FD-PRP in the oral and maxillofacial region that would primarily assess safety. Furthermore, two kinds of grafting materials with different characteristics, β-TCP and carbonate apatite, were used. In addition, there were two surgical approaches: staged and simultaneous. Therefore, comparative studies with larger samples that consist of a unified grafting material and surgical approach performed with or without FD-PRP are required to validate our findings about the efficacy of applying x3FD-PRP for bone regeneration. Secondly, the preparation of PRP was carried out manually according to our previous study; therefore, constant bias may occur between coordinators. We should use the standardized technique, such as utilizing a PRP purification kit, in our future study. Thirdly, scientific evidence does not strongly support the use of PRP due to the lack of controlled clinical trials. There are several studies that do not support the synergy of PRP or platelet-rich fibrin for bone regeneration [17,18,19]. Future research in this field should be directed toward the implementation of well-designed, adequately powered, RCTs.

In conclusion, the results obtained in this study suggest that x3FD-PRP, for up to 4 weeks after freeze-dried storage, can be used safely for bone engineering in clinical practice, although there are several limitations in the present study, especially the small sample size. Further studies are required to determine both the safety and efficacy of x3FD-PRP.

## Figures and Tables

**Figure 1 jcm-10-01678-f001:**
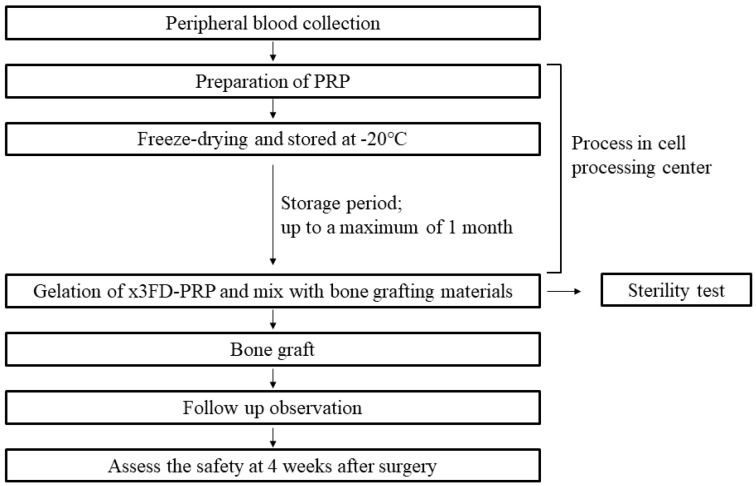
Protocol of this study. All processes of FD-PRP preparation were performed in a sterile room at the Nagasaki University hospital cell processing center, and the surgery was performed as a clean operation. A sterility test was performed in part of FD-PRP which was used for transplant. The safety verification was evaluated in terms of the clinical course until 4 weeks after surgery.

**Figure 2 jcm-10-01678-f002:**
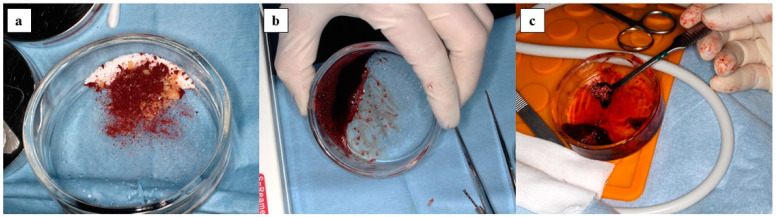
The preparation process for the complex of x3FD-PRP and bone grafting materials is shown: (**a**) FD-PRP powder and white particulate bone grafting materials; (**b**) complex of gelated x3FD-PRP and bone grafting materials; (**c**) given good operability.

**Figure 3 jcm-10-01678-f003:**
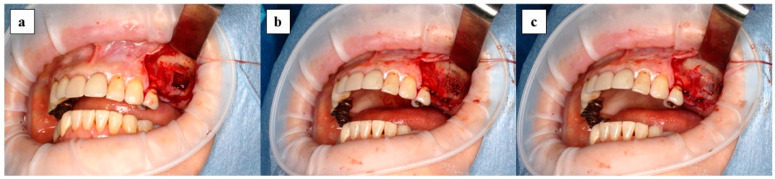
Surgical procedure of maxillary sinus floor augmentation is shown: (**a**) bone window is removed and the maxillary sinus mucous membrane is elevated; (**b**) complex of x3FD-PRP and bone grafting materials are transplanted; (**c**) bone window is repositioned.

**Figure 4 jcm-10-01678-f004:**
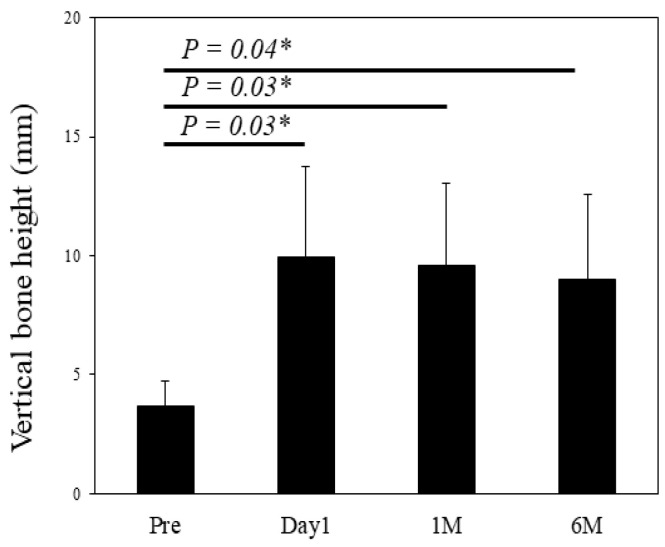
Quantitative measurement of vertical bone height at the transplanted site is shown. *; *p* < 0.05, statistically significant difference.

**Figure 5 jcm-10-01678-f005:**
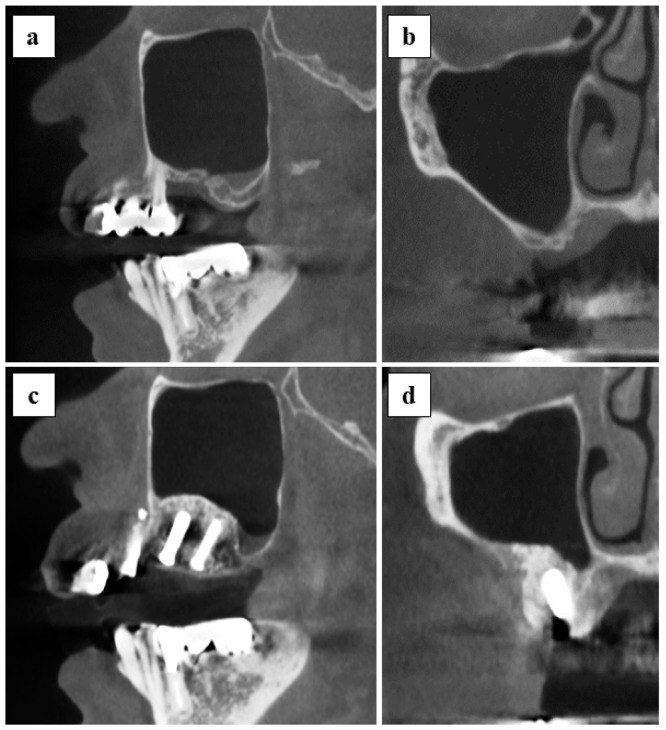
CT findings of pre-operative and post-operative at 6 months are shown: (**a**,**b**) pre-operative images, sagittal section and coronal section; (**c**,**d**) post-operative 6 months images, sagittal, and coronal section.

**Table 1 jcm-10-01678-t001:** List of the patients who participated in the clinical trial.

Case	Gender	Age	Surgical Procedures	Implant Placement Site	Amount of x3FD-PRP	Bone Grafting Materials
1	F	39	Lt. sinus lift, GBR	24	2 mL	Cerasorb M^®^
2	F	64	Lt. sinus lift	25, 26	1 mL	Cerasorb M^®^
3	F	59	Rt. sinus lift, GBR	15, 16	2 mL	Cytrans Granules^®^
4	M	41	Rt. sinus lift	staged approach	1 mL	Cytrans Granules^®^
5	F	57	Lt. sinus lift	staged approach	2 mL	Cytrans Granules^®^

Lt; left side. Rt; right side. GBR; guided bone regeneration.

**Table 2 jcm-10-01678-t002:** Blood cell counts in peripheral blood and platelet-rich plasma.

	PB	PRP	*p* Value
WBC (×10^2^)	46.4 (8.1)	148.6 (45.6)	0.008 *
RBC (×10^4^)	421.4 (71.1)	391.4 (101.1)	0.37
Platelet (×10^4^)	17.4 (3.9)	94.7 (20.5)	0.001 *

Data are presented as mean (SD). WBC; white blood cell. RBC; red blood cell. PB; peripheral blood. PRP; platelet-rich plasma. *; *p* < 0.05, statistically significant difference.

**Table 3 jcm-10-01678-t003:** ISQ value.

Case	ISQ Value (Implant Placement Site)
1 2 3	80 (24) 77 (25), 70 (26) 71 (15), 70 (16)
Mean ± SD	73.6 ± 4.1

## Data Availability

The data presented in this study are available on request from the corresponding author.

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
