# Peer review of "Clinical Safety Assessment of Autologous Freeze-Drying Platelet-Rich Plasma for Bone Regeneration in Maxillary Sinus Floor Augmentation: A Pilot Study"

_jcm, 2021, doi:10.3390/jcm10081678_

Round 1

Reviewer 1 Report

I think the Authors have investigated an interesting aspect.
If the non-transfusional frized blood product proved to be equally effective, the surgical procedures would be greatly simplified.
It should be noted that in some Countries, such as mine, however, such storage would be illegal if carried out by the dentist and it should be done by a MD specialized in immunotransfusional medicine.

Unfortunately, the method used for the analysis certifies the safety of the product but not its effectiveness, for this reason I would suggest changing the text and also the title of the research.

page 2 line 54

"but they have limited osteoinductivity"

I disagree: they may have limited or NONE osteoinductivity

page 2 line 82

Bone grafting materials are two: this is not a real single assignment study...

page 5 line 174

You may cite this paper in order to describe this procedure:

Bacci C, Sivolella S, Brunello G, Stellini E. Maxillary sinus bone lid with pedicled bone flap for foreign body removal: the piezoelectric device. Br J Oral Maxillofac Surg. 2014 Dec;52(10):987-9. doi: 10.1016/j.bjoms.2014.04.019. Epub 2014 Jun 2. PMID: 24894711.

page 8 line 272

"...new bone"

The data presented does not allow this statement

Some choices, such as using two different bone substitute materials in such a small sample, should be discussed.

Author Response

We are grateful to your valuable suggestions, which make us improve our manuscript.

I think the Authors have investigated an interesting aspect.
If the non-transfusional frized blood product proved to be equally effective, the surgical procedures would be greatly simplified.

Thank you for the comment.

It should be noted that in some Countries, such as mine, however, such storage would be illegal if carried out by the dentist and it should be done by a MD specialized in immunotransfusional medicine.

We are sorry for the situation, but we believe the issue will be resolved when a specific system for preparation and delivery pf FD-PRP will be established. We believe the organization of the system is necessary to spread widely in clinical practice.

Unfortunately, the method used for the analysis certifies the safety of the product but not its effectiveness, for this reason I would suggest changing the text and also the title of the research.

Thank you for an important suggestion we have revised as the followings.

  1. title as “Clinical safety assessment…..”. (P1L2)
  2. Abstract: to evaluate the safety and preliminary efficacy….(P1L20)

page 2 line 54

"but they have limited osteoinductivity"

I disagree: they may have limited or NONE osteoinductivity

Yes, we agree, then, have revised as “don’t have osteoinductivity” (P2L58)

page 2 line 82

Bone grafting materials are two: this is not a real single assignment study...

Thank you for your indication. However, the subject of the assessment in this clinical study is FD-PRP but not grafting materials. Then, this study is a single assignment study.

page 5 line 174

You may cite this paper in order to describe this procedure:

Bacci C, Sivolella S, Brunello G, Stellini E. Maxillary sinus bone lid with pedicled bone flap for foreign body removal: the piezoelectric device. Br J Oral Maxillofac Surg. 2014 Dec;52(10):987-9. doi: 10.1016/j.bjoms.2014.04.019. Epub 2014 Jun 2. PMID: 24894711.

We don’t think the suggested citation is necessary because the paper describes a foreign body removal but not maxillary sinus floor augmentation, and the use of piezoelectric device for maxillary sinus floor augmentation has already been popular now.

page 8 line 272

"...new bone"

The data presented does not allow this statement

According to your suggestion, we revised as “Our experimental results may indicate that x3FD-PRP could be applied safely and shows no adverse effects for maxillary sinus floor augmentation.” (P8L281)

Some choices, such as using two different bone substitute materials in such a small sample, should be discussed.

Yes, we used two kinds of grafting materials. It is mostly because no synthetic bone substitute including Cerasorb had not been officially approved to use implant dentistry in Japan at the beginning of this study, but later on Cytrans Granules had been approved to use for implant dentistry. However, we recognize that the material should be unified.

The, we have revised and added the following sentences in Discussion. “First, the most critical impediment of this study is a small sample size including five subjects with only one male with a single arm, even though this was planned as a first-in-human study to apply x3FD-PRP in oral and maxillofacial region to assess the safety primarily. Furthermore, two kinds of grafting materials, β-TCP or carbonate apatite, which have different characteristics, were used, and also there were two surgical approaches, staged or simultaneous. Therefore, comparative studies with larger samples, consisting of unified grafting material and surgical approach with or without FD-PRP, are required to validate our findings about the efficacy of x3FD-PRP application for bone regeneration.” (P9L342)

Reviewer 2 Report

Clinical safety and efficacy of autologous freeze-drying platelet-rich plasma for bone regeneration in maxillary sinus floor augmentation: A preliminary pilot study

Thank you for the opportunity to review this paper. I think the subject area will be interesting to readers and is likely to represent the future of bone augmentation in the maxillofacial region.

Typo Page 1 Para 1, Line 47 last sentence – Therefor – Therefore

Page 2, Line 51 Missing ‘the’, page 2 grafting is still the gold standard

Page 2, Line 76  - Missing ‘the’ under 2.1 registered in the Japan Registry

The title of what is referred to as the current study is noted to be - “Efficacy of freeze-dried platelet-rich plasma (FD-PRP) in bone engineering: Pilot study in bone engineering” . Which differs from the title of the paper. Does this matter?

Page 4, Line 150 – ‘L’ missing from Lyophilization?

Page 4, Line 172 might be better to be pushed to page 5?

Page 5, Line 173 – should thorough – read as through?

Page 8, line 287 extra ‘the’ ….support that the safety  is secured…

Page 9, Line 329 refers to preliminary study, while earlier references are to a pilot study. There is a difference between the two. I think should clarify which one. Wouldn’t it be true that, if you plan a different methodology this could be considered a preliminary study? It’s registered as a pilot though? The title too is then a little confused.

Page 9, Line 336 the beginning of the sentence is grammatically incorrect? Should it read there are several studies which do not support…?

The main limitations revolve around this being a study of just 5 individuals, with just 1 male., and with heterogeneous surgery (staged, +/- GBR) and so the results are difficult to apply.  When the safety verification assessment is completed some of the outcomes are expected in any surgery – ie, in cases when no grafting with or without PRP is undertaken. Similarly as it is only 5 cases these outcomes could be absent – in cases where grafting with or without PRP was completed…?

Is there any reason that a control using xenograft alone couldn’t have been completed?

Would be interesting to know whether the scheme for the panoramic  and  CT tomography Page 5, Line 197 etc is routine practice in these cases or additional for the study?  Maybe of little value but if the xray scheme post-op is over and above routine it could be seen as inappropriate additional radiation?

I would be inclined to be a little more descriptive in the conclusions about the limitations of the study. 

Author Response

We appreciate you for reviewing our manuscript intensively and giving us helpful suggestions and comments. Following those suggestions, we revised our manuscript with best of our efforts.

Thank you for the opportunity to review this paper. I think the subject area will be interesting to readers and is likely to represent the future of bone augmentation in the maxillofacial region.

Thank you for your kind comment.

Typo Page 1 Para 1, Line 47 last sentence – Therefor – Therefore

Page 2, Line 51 Missing ‘the’, page 2 grafting is still the gold standard

Page 2, Line 76  - Missing ‘the’ under 2.1 registered in the Japan Registry

Page 4, Line 150 – ‘L’ missing from Lyophilization?

Page 4, Line 172 might be better to be pushed to page 5?

Page 5, Line 173 – should thorough – read as through?

Page 8, line 287 extra ‘the’ ….support that the safety  is secured…

Thank you for your indications. We revised all according to your suggestions.

The title of what is referred to as the current study is noted to be - “Efficacy of freeze-dried platelet-rich plasma (FD-PRP) in bone engineering: Pilot study in bone engineering” . Which differs from the title of the paper. Does this matter?

We believe this doesn’t matter. The referred study title was officially registered before an initiation of the study, but we recognized this title do not fit the results of this clinical study. Then, we have changed the title for the paper.

Page 9, Line 329 refers to preliminary study, while earlier references are to a pilot study. There is a difference between the two. I think should clarify which one. Wouldn’t it be true that, if you plan a different methodology this could be considered a preliminary study? It’s registered as a pilot though? The title too is then a little confused.

We appreciate your important indication. We confused the way to use “pilot” and “preliminary”, and we unified to use “pilot” through the manuscript including the title.

Page 9, Line 336 the beginning of the sentence is grammatically incorrect? Should it read there are several studies which do not support…?

Thank you for the indication. We have revised the sentence as “Third, PRP has not been strongly supported by scientific evidence due to lack of controlled clinical trials.” (P9L353)

The main limitations revolve around this being a study of just 5 individuals, with just 1 male., and with heterogeneous surgery (staged, +/- GBR) and so the results are difficult to apply.  When the safety verification assessment is completed some of the outcomes are expected in any surgery – ie, in cases when no grafting with or without PRP is undertaken. Similarly as it is only 5 cases these outcomes could be absent – in cases where grafting with or without PRP was completed…?

Yes, you are completely right. When we planed this clinical study, there were no clinical studies to apply lyophilized PRP in humans. Then, we intended to conduct this study as a first-in-human study to assess the safety of x3FD-PRP primarily with a small number of subjects following RCT with a larger number of subjects. However, we could not succeed in recruiting subjects so that it took long time to carry out the study. Now, we are planning to conduct a RCT with enough funding support to perform a clinical study.

We have revised the discussion concerning the limitation of this study intensively as the following, “First, the most critical impediment of this study is a small sample size including five subjects with only one male with a single arm, even though this was planned as a first-in-human study to apply x3FD-PRP in oral and maxillofacial region to assess the safety primarily. Furthermore, two kinds of grafting materials, β-TCP or carbonate apatite,which have different characteristics, were used, and also there were two surgical approaches, staged or simultaneous. Therefore, comparative studies with larger samples, consisting of unified grafting material and surgical approach with or without FD-PRP, are required to validate our findings about the efficacy of x3FD-PRP application for bone regeneration.” (P9L342)

Is there any reason that a control using xenograft alone couldn’t have been completed?

We recognize that xenograft, especially BioOss, is most popular (or standard) grafting material for maxillaru sinus augmentation in the world, but Japanese government has approved it to use periodontal defects but not to use in implant dentistry. Then, we do not use the material. As mentioned above, we think we need to conduct a RCT.

Would be interesting to know whether the scheme for the panoramic  and  CT tomography Page 5, Line 197 etc is routine practice in these cases or additional for the study?  Maybe of little value but if the xray scheme post-op is over and above routine it could be seen as inappropriate additional radiation?

We routinely take a panoramic x-ray at day 1 and about 6 month after surgery, and CT at about 6 month after surgery for staged approach. But the other X-rays were took additionally to evaluate the augmented conditions. As you indicated, there is a risk to suffer from additional radiation, but we believe the information taken from this assessment overcome the risk. IRB approved the protocol, and we explained the subjects concerning the risk of additional radiation and got the informed consent.

I would be inclined to be a little more descriptive in the conclusions about the limitations of the study. 

Thank you for your suggestion. We have added the following sentence in conclusion, “although there are several limitations, especially a small sample size, in the present study.” (P9L359)

Round 2

Reviewer 1 Report

Paper has been improved